# Chromosome-Scale Genome Assembly of the Sheep-Biting Louse *Bovicola ovis* Using Nanopore Sequencing Data and Pore-C Analysis

**DOI:** 10.3390/ijms25147824

**Published:** 2024-07-17

**Authors:** Chian Teng Ong, Karishma T. Mody, Antonino S. Cavallaro, Yakun Yan, Loan T. Nguyen, Renfu Shao, Neena Mitter, Timothy J. Mahony, Elizabeth M. Ross

**Affiliations:** 1Centre for Animal Science, Queensland Alliance for Agriculture and Food Innovation, The University of Queensland, Brisbane, QLD 4072, Australia; chianteng.ong@uq.edu.au (C.T.O.); a.cavallaro@uq.edu.au (A.S.C.); yakun.yan@uq.net.au (Y.Y.); t.nguyen3@uq.edu.au (L.T.N.); n.mitter@uq.edu.au (N.M.); t.mahony@uq.edu.au (T.J.M.); 2Centre for Bioinnovation, University of the Sunshine Coast, Sippy Downs, QLD 4556, Australia; rshao@usc.edu.au; 3School of Science, Technology and Engineering, University of the Sunshine Coast, Sippy Downs, QLD 4556, Australia

**Keywords:** *Bovicola ovis*, Pore-C, genome assembly

## Abstract

*Bovicola ovis*, commonly known as the sheep-biting louse, is an ectoparasite that adversely affects the sheep industry. Sheep louse infestation lowers the quality of products, including wool and leather, causing a loss of approximately AUD 123M per annum in Australia alone. The lack of a high-quality genome assembly for the sheep-biting louse, as well as any closely related livestock lice, has hindered the development of louse research and management control tools. In this study, we present the assembly of *B. ovis* with a genome size of ~123 Mbp based on a nanopore long-read sequencing library and Illumina RNA sequencing, complemented with a chromosome-level scaffolding using the Pore-C multiway chromatin contact dataset. Combining multiple alignment and gene prediction tools, a comprehensive annotation on the assembled *B. ovis* genome was conducted and recalled 11,810 genes as well as other genomic features including orf, ssr, rRNA and tRNA. A manual curation using alignment with the available closely related louse species, *Pediculus humanus*, increased the number of annotated genes to 16,024. Overall, this study reported critical genetic resources and biological insights for the advancement of sheep louse research and the development of sustainable control strategies in the sheep industry.

## 1. Introduction

In Australia, sheep flocks are farmed mainly for meat and wool, with Australian wool and meat production estimated at AUD 3.6B and 4.3B respectively. *Bovicola ovis*, (formerly *Damalinia ovis*), also commonly known as the sheep-biting louse, is an ectoparasite that lives and feeds on dead skin, secretions and bacteria normally found on sheep surface. *Bovicola ovis* is a 1.5 to 2 mm long pale-yellow insect that infests the skin of sheep at the neck, shoulder, sides and back [1]. Host-specific [2] lice infest livestock creating an adverse economic impact on agriculture. *Bovicola ovis* causes one of the top infestations that economically affect the Australian sheep industry. Infestation of *B. ovis* in sheep triggers the hosts’ pruritic responses, including rubbing, biting and scratching, causing a reduction in wool quality and yield by up to 1 kg per sheep in a single year [3,4,5]. In Australia alone, it is projected that 23.3% of sheep flocks will be afflicted by sheep lice infestations, which subsequently results in production losses of approximately AUD 123M per annum. Furthermore, farmers will incur additional costs associated with treating the affected sheep [6,7,8].

Sheep lice are primarily diagnosed by wool parting and observation; however, its effectiveness is subject to the inspector’s skills and experiences and is extremely time-consuming with a larger flock [9]. A contributing factor to the challenges associated with sheep lice accurate detection is the small size (2 mm) of the parasite making them difficult to identify. It has been reported that infestations of less than 400 sheep body lice often go unnoticed [10]. Currently, the only control method for sheep lice control is treating sheep with chemical lousicides. However, resistance, lack of specificity, improper use and continual reliance on chemicals make the long-term management of the parasite increasingly problematic for the Australian sheep industry [10]. The rapid emergence of resistance against many insecticides and the increased prevalence of treatment-resistant lice populations pose a challenge to the continued control of sheep lice. Additionally, the potential presence of chemical residues in wool scouring effluent and occupational exposure to chemicals has led to an urgent need for alternative methods for the sustainable control of sheep lice [11,12]. Some alternative control measures like vaccination and compounds like tea tree oil have been evaluated, but no effective solution is yet available against sheep lice [13].

There have been several attempts to improve the detection and diagnosis of sheep lice infestations. For example, an enzyme-linked immunosorbent assay (ELISA) with highly sensitive visual detection was developed and was used commercially as an “on-farm” diagnostic tool [14]. However, the *B. ovis* ELISA detection assay required a minimum 3-day processing time, costing AUD 134 per sample. Therefore the field application was eventually terminated due to limited utilisation by the sheep producers who preferred the less time-consuming and more cost-effective method of visual identification despite its limited sensitivity [6,14]. Since 2020, various molecular methods, including Polymerase Chain Reaction (PCR), looped-mediated isothermal amplification (LAMP) and quantitative PCR (qPCR), have been used for sheep louse detection [7,15]. A LAMP assay demonstrated a higher percentage of sensitivity and specificity (100% and 75%, respectively) than PCR (80% and 33%, respectively) when crude DNA was used for sheep louse detection, potentially because of the non-specific amplification of target genes posed by other species in the crude DNA extracted by direct boiling of the residues of cutters and combs that were used for shearing [7]. The sensitivity and specificity estimate of sheep lice detection using the PCR assay were improved to 100% when DNA was extracted with a commercial kit. However, using the crude DNA extract and LAMP assay was considered the preferred choice because this combination was less laborious and more time-efficient for field applications. However, this study utilised template DNAs from using 80 mg pooled samples and therefore may only be useful for managing lice infestations at the flock level [8]. It is likely that a sensitive and specific detection at the individual animal level is required for long-term and sustainable control detection of lice infestations.

Another study explored the capability of molecular tests to call out individual infestations [15]. This study reported improved detection power to confidently detect low levels of *B. ovis* DNA down to 5 × 10^−8^ and 5 × 10^−6^ ng/μL for qPCR and LAMP assays, respectively. Additionally, this approach allowed flexible wool collection times and shortened the molecular assay processing time to less than an hour for multiple samples. A field-appropriate fleece swabbing method was developed in the study to replace the original fleece dissolving method, which decreased the sample processing time by 2 h. Although the sensitivity was decreased slightly, when combined, fleece sampling by swabbing and the LAMP assay provided a high-throughput and time-efficient tool for *B. ovis* detection [15].

The requirement to find more sustainable, safer, pest-specific alternatives to enable improved control, or potential to eradicate, sheep lice remain an unresolved challenge that requires a new approach. One such novel approach for sheep lice control is the application of RNA interference (RNAi) technology. RNAi has demonstrated great promise in combating various pests and diseases including those that affect livestock [16]. RNAi uses short RNA molecules to suppress the expression of genes critical for pests/pathogens. The availability of genomic and transcriptomic information on sheep lice can provide researchers with the opportunity to potentially develop a sustainable species-specific biological control against sheep lice [17]. For example, malaria research accelerated with respect to both diagnostic and disease management when the first reference genome assemblies were published for the malaria parasite, *Plasmodium falciparum*, and its most important mosquito vector, *Anopheles gambiae* [18]. Due to the detailed genomic data provided by reference genomes, breakthroughs were achieved in understanding the pathology, epidemiology, drug resistance, vaccine development and disease control [19,20,21].

The aim of this study was the establishment of a reference genome for *B. ovis* to aid in the development of new biological-based strategies for the prevention and control of lice infestation of sheep. By effectively combining genome sequence data and chromatin conformation information, this research culminated in a meticulously curated and well-annotated *B. ovis* genome derived using mixed-life stages of *B. ovis* (nymph, female and male). Notably, this study imparts valuable insights for an array of subsequent inquiries, encompassing not only the fundamental biology of lice but also their epidemiology. Furthermore, this comprehensive genome dataset holds significant promise for pest diagnostic and treatment investigations, which hold substantial potential to enhance the well-being of livestock industries.

## 2. Results

In total, 50 µL of 92 ng/µL of high molecular weight whole genomic DNA was extracted from the pooled sheep lice sample. Ideally, the reference genome of an organism should be derived from a single cell line or individual in order to avoid technical issues with genetic polymorphisms. However, due to the lack of *B. ovis* cell lines and the requirement of high molecular weight DNA as the starting material for sequencing, a pool of sheep lice *B. ovis* was collected for DNA extraction. Approximately 372 Gb of raw sequencing signal data was generated, which resulted in 38.39 Gb of sequence (7.94 million reads). The average N50 of the sequenced reads was 16.4 kb. After filtering out reads that were below 500 bp or had a quality score < 12, as well as reads that mapped to either the prokaryotic or the ovine assemblies, 8.53 Gb bases (~2.23 million reads) were retained for the assembly of the *B. ovis* genome. GenomeScope 2.0 estimated a haploid genome size of 227.39 Mbp, heterozygosity of 0.62% and 1.62 error rate based on the K-mer distribution on the filtered reads (Figure 1A). After removing duplicated contigs and assembly polishing, the *B. ovis* genome comprised 788 contigs with a total size of 150,622,374 bp, with a raw read coverage of ×54.

To improve the final assembly, scaffolding of the decontaminated draft assembly using the amplification-free Pore-C data (Figure 1B) and careful curation were performed. The final *B. ovis* genome comprises 320 scaffolds, with a total genome size of 123,066,105 bp, scaffold N50 of 624 Kb and BUSCO score 96.6% (Figure 2). The largest scaffold was 4.67 Mb in size while the smallest scaffold was 26 Kb. The final *B. ovis* genome assembly in this study was ~123 Mbp in size, which is consistent with other louse genomes, including *P. humanus* (~110 Mbp), *Brueelia nebulosa* (~114 Mbp), *Ricinus arcuatus* (~155 Mbp) and *Columbicola columbae* (~208 Mbp) (Table 1). The BUSCO completeness score remained at 96.6% after scaffolding; however, the number of scaffolds was reduced by approximately 60% from 788 to 320 scaffolds.

The low-complexity genome regions and interspersed repeats were identified by homologous comparison using RepeatMasker by searching against the Repbase database of *A. gambiae* and *Drosophila.* Approximately 0.05% of interspersed repeats and 11.17% of simple repeats were identified. The repetitive content of *B. ovis* assembly generated in this study had a higher repetitive element content than *P. humanus* (~7%) [24] and *C. columbae* (~9%) [23], but a smaller number of repetitive elements than *B. nebulosa* (~15%) [22]. A masked consensus was created from the RepeatMasker results.

Evidence-based alignment using either the published *P. humanus corporis* (GCF_000006295.1), or general dataset including RefSeq [25] and SWISS-PROT [26] was performed to train the gene prediction tools and to provide guidance for manual structural curation. In general, alignments against the RefSeq database reported a higher number of matches compared to alignments against the *P. humanus corporis* (GCF_000006295.1), regardless of whether transcripts or polypeptides were used (Table 2). Using the *P. humanus corporis* data, the transcript-based alignment using blastn 2.12.0 [27] and BLAT v2.5 [28] returned 3801 and 241 matched, respectively, while the protein-based alignment using blastx [27] and DIAMOND [29] returned 7253 and 15,863 matches, respectively. In comparison, alignment against the RefSeq invertebrate database reported 110,523 matches with blastn [27], 331,625 matches with BLAT [28] and 6,043,377 matches with the DIAMOND alignment tool [29].

The reference-based structural gene prediction using AUGUSTUS 3.4.0 [30] based on the *D. melanogaster* predefined training set only identified 1612 genes, while it identified 20,977 genes based on the customised training set of *P. humanus corporis* (GCF_000006295.1) (Table 3). Using the alignment results generated by HISAT2 2.2.1 [31] and Tophat 2.1.1 [32], BRAKER2 2.1.5 pipeline [33] predicted 11,184 genes, with an average gene length of 4874 bp, from the 85,358 exons and 73,024 introns identified. The annotation was estimated to be 94.7% complete based on the arthropoda_odb10 dataset by BUSCO 5.2.2 [34]. Ab initio prediction with GeneMarkES 4.48 [35] predicted 11,313 genes from the unsupervised training set and achieved 93.8% completeness by BUSCO 5.2.2 [34] on the arthropoda_odb10 dataset. SNAP 11/29/2013 [36] found only 1438 genes based on the *D. melanogaster* pre-trained HMM dataset. Other genomic structures, including the open reading frames, simple sequence repeat (SSR) markers, ribosomal RNA (rRNA) and transfer RNA (tRNA) in the masked consensus were annotated (Table 4).

To obtain a comprehensive gene structure annotation, an official gene set for the *B. ovis* assembly was produced using EvidenceModeler 1.1.1 [37] by combining the differently weighted functional annotations from the sequence homologies and the structural annotation result generated by the gene prediction tools. PASA [38] was employed to weigh and refine the official predicted gene set using the cDNA transcripts of *P. humanus corporis* (GCF_000006295.1) (Table 5), which resulted in 11,810 genes, 11,810 mRNA and 11,444 proteins in total. The official gene set (*n* = 11,810) was functionally annotated using various tools, including the *P. humanus corporis* (GCF_000006295.1) protein sequences, RefSeq invertebrate [25] and SwissProt [26], using different functional annotation platforms (blastp [27], DIAMOND [29], InterProScan 5.53 [39], Pfam 1.6 [40] and SignalP 5.0 [41] (Table 6). The number of genes identified from the *B. ovis* assembly in the current study was comparable to that of *P. humanus* (10,993 genes) and *B. nebulosa* (10,587 genes) [22,24]. Manual gene curation using blastn transcript alignment against *P. humanus* (GCF_000006295.1) and RefSeq invertebrate database increased the number of annotated genes to 16,024.

The *B. ovis* sequenced in this study and the *P. humanus corporis* (GCF_000006295.1) shared 99.1% and 98.4% of their orthogroups with each other, respectively. There were 541 genes (4.6%), which were grouped in 121 orthogroups, uniquely expressed in *B. ovis*. In total, only 195 of the 541 uniquely expressed genes in *B. ovis* were assigned to a COG functional group, of which 71 (36.4%) were labelled as “Function unknown” (Appendix A). The most abundant COG group with known function was “Replication and repair” (*n* = 22), followed by and “Cytoskeleton” (*n* = 21), “Signal Transduction” (*n* = 19) (Figure 3). Additionally, 167 of the unique genes returned no blast results and 170 were labelled as either hypothetical, unnamed proteins or uncharacterized proteins (Appendix A). 

## 3. Discussion

Here, we present the first genome assembly for the sheep louse (*B. ovis*). We incorporated innovative approaches, including third-generation sequencing and Pore-C analysis, to generate a high-quality reference assembly and annotation for *B. ovis*, a pernicious parasite affecting Australian sheep industries. The incorporation of third-generation sequencing enabled a highly continuous de novo genome assembly to be completed. The target organism in this study, *B. ovis*, previously had no assembled and annotated genome, nor was there any genome assembly from any organism within the same genus. The closest related published genome, *Pediculus humanus*, which was the human head louse, also belongs to the same taxonomical Order of Phthiraptera as *B. ovis*. The significance of this research work on the sheep lice genome lies in the fact that it will serve as a valuable reference for other animal lice genomes, as currently there is limited available information on them.

The high BUSCO completeness score and low number of contributing scaffolds achieved here demonstrated the advantages of Pore-C scaffolding for whole-genome assembly by capturing the proximal interactions, which were used to correct and merge the contigs. Therefore, the fragmentation of the assembled genome was minimized, and the chromosome-level assembly was achieved. An increasing number of studies [42,43,44] have incorporated three-dimensional chromatin conformation analysis, including Hi-C and Pore-C, in a whole-genome assembly study to improve the scale of “traditional” phased linear de novo assembly.

In this study, we reported a chromosome-scale genome assembly of the sheep-biting louse, *B. ovis*, using Nanopore sequencing data and Pore-C analysis. Pore-C captures a comparable number of chromosomal interactions to Hi-C with less sequencing data, demonstrating high efficiency [45,46,47]. Our chromatin conformation analysis showed 58.17% two-way interactions and 41.83% multi-way interactions, similar to human cells (47.56%) [47] and Arabidopsis (44%) [46]. Longer reads with more contacts provide richer multi-dimensional interaction data, enhancing the quality of de novo genome assembly. Although only one Pore-C library was generated due to limited tissue, the high consistency in Pore-C data from Arabidopsis [46], suggests the data here are of sufficient quality for assembly. This highlights Pore-C’s reliability and effectiveness, even with limited samples, justifying its use in complex genomic studies.

Our analysis revealed a relatively high similarity between *B. ovis* and *P. humanus corporis* (GCF_000006295.1) as 99.1% of the gene orthologs in *B. ovis* were also reported in *P. humanus corporis* (GCF_000006295.1). Further investigation of the unique genes in *B. ovis* reported several key genes that may be linked to pesticide resistance mechanisms in *B. ovis*. For instance, two cytochrome b-c1 complex subunit 8-like genes were identified, which are crucial for cellular respiration and energy production. These cytochromes might play significant roles in metabolic adaptations and resistance [48]. Several other genes that can have potential roles in resistance mechanisms have been identified: an ABC transporter (putative), known for mediating multidrug resistance [49] and acetylcholine receptor protein subunit alpha-L1 precurso(putative), potentially linked to neurotoxin resistance [50]; a dual specificity protein phosphatase CDC14 (putative) involved in cell cycle regulation [51]; and glucose dehydrogenase (putative) impacting energy metabolism [52]. Furthermore, identified genes like the ionotropic glutamate receptor, histone-lysine N-methyltransferase SETMAR, and Rho-associated protein kinase can also further our understanding of resistance to insecticides. Overall, these findings provide valuable insights into the molecular mechanisms of *B. ovis* resistance against control treatments, aiding the development of effective control strategies for the sheep industry.

In conclusion, our study described the assembly of the first *B. ovis* sheep lice genome. The assembly improvement with the amplification-free multiway chromatin conformation analysis and the comprehensive gene annotation in chromosome-level scaffolds. Since this is the first high-quality genome of ectoparasitic lice that affect livestock, the assembled genome and the genetic characteristics can be referred to as fundamental resources for lice science and inform the advancement of infection management and control, including accurate diagnostic methods and more effective prevention measures in livestock industries. For example, an accurate genome sequence could inform the development of sequence-specific treatments, such as those that utilise RNAi technology to target genes that are essential for sheep lice survival without adversely affecting host or other non-target organisms [53]. Another potential diagnostic approach using the *B. ovis* genome is adaptive sequencing for specific and accurate detection as demonstrated in a previous study [54]. With the advancement of technologies, a high-quality pest reference genome provides the basis for the future development of time-efficient and cost-effective control strategies against livestock pest diseases, which are crucial to both animal welfare and livestock production.

## 4. Materials and Methods

### 4.1. Sheep Louse Sampling

Wool samples from sheep suspected of louse infestation were generously provided by local Queensland sheep farms. A piece of black background cloth was laid out, and a warm lamp was set up above it. The sheep wool was illuminated using the warm lamp, causing the lice to fall onto the black cloth. Additionally, the wool was spread out on the black background cloth, and lice that were firmly attached to the wool were removed with tweezers. All collected samples were then placed into 1.5 mL Eppendorf tubes (Eppendorf, Hamburg, Germany) for the next steps.

### 4.2. Sheep Louse RNA Extraction

To extract the RNA, approximately 25 mg of sheep lice were added to TRIsure (Bioline, London, UK) prior to homogenisation using TissueLyser II (QIAGEN, Hilden, Germany). Chloroform was added for chemical cell lysis. After incubation and centrifugation, the aqueous phase of RNA was separated, and isopropyl alcohol was added for RNA precipitation. The precipitated RNA was purified with 70% ethanol and treated with DNaseI. The concentration and quality of the RNA were assessed with Nanodrop and agarose gel.

### 4.3. Sheep Louse High Molecular Weight Genomic DNA Extraction

Cryogenic homogenization was conducted by pulverising sheep lice with a mortar and pestle. The genomic DNA (gDNA) was extracted from approximately 25 mg of the ground sheep lice tissue using Puregene Tissue kit (QIAGEN, Hilden, Germany) by following the manufacturer’s instructions. Briefly, the ground tissue was treated with proteinase K and RNase A before protein precipitation. Isopropanol was added to the supernatant for gDNA precipitation. The precipitated gDNA was purified and stored at −20 °C. The quality and length of the extracted gDNA were observed with pulsed-field gel electrophoresis (Pippin Pulse, Sage Science, MA, USA) and the quantity of the gDNA was measured using Qubit^TM^ 4 fluorometer (Thermo Scientific, DE, USA).

### 4.4. Sheep Louse Chromatin Cross-Linked DNA Extraction

Another ~25 mg of cryo-ground sheep louse tissue was added to 1 mL of pre-chilled phosphate-buffered saline (PBS). Pore-C DNA extraction was conducted as described previously [42]. Briefly, the tissue pellet was resuspended with 1% formaldehyde solution and kept at room temperature for 10 min for cross-linking. The cross-linked sample pellet was treated with protease inhibitor cocktail-permeabilization solution for cell lysis. The suspended pellet was gently mixed with pre-chilled 1.5× digestion reaction buffer, followed by incubation at 65 °C with 1% sodium dodecyl sulphate (SDS), which was quenched by 10% ECOSURF EH-9 later, for chromatin denaturation. Restriction enzyme NlaIII was added to digest the chromatin. The restriction enzyme digestion was subsequently inactivated before the proximity ligation reaction (5% Tween-20, 0.5% SDS and proteinase K) was added to the mixture. Protein degradation reaction was added to the mixture for an incubation period of 18 h at 56 °C to reserve cross-links and degrade proteins. Pore-C DNA extraction DNA extraction was performed with pre-chilled phenol:chloroform:isoamyl alcohol 25:24:1. The pelleted cross-linked DNA was purified with 70% ethanol prior to precipitation.

### 4.5. Sheep Louse Transcriptomic Sequencing and Data Processing

Before preparing RNA-Seq libraries for sequencing, RNA samples were quantified using the Qubit™ RNA broad-range Assay Kit (Invitrogen, Carlsbad, CA, USA), and QC was performed using the Agilent RNA tapes (#5067-5576) on the TapeStation 4200 (Agilent # G2991AA) as per the manufacturer’s instructions. Total RNA (30 ng) input was used to prepare RNA-Seq libraries using Illumina Stranded Total RNA Prep with Ribo-Zero Plus (Illumina #20040529) as per the manufacturer’s instructions. The only alteration to the protocol was the reduction in the total reaction volume from XX µL to YY µL to accommodate samples with lower concentrations and/or volumes. According to the library preparation protocol and depending on the input RNA amount, 17 PCR cycles were performed to amplify and add indexes and primer sequences in preparation for sequencing. On completion of the library preparation protocol, each library was quantified, and QC was performed using the Quant-iT™ dsDNA HS Assay Kit (Invitrogen) and Agilent D1000 HS tapes (#5067-5582) on the TapeStation 4200 (Agilent # G2991AA) as per the manufacturer’s protocol.

RNA-Seq libraries were pooled at equimolar amounts of 2 nM per library to create a sequencing pool. The library pool was quantified in triplicates using the Qubit™ dsDNA HS Assay Kit (Invitrogen). Library QC is performed using the Agilent D1000 HS tapes (#5067-5582) on the TapeStation 4200 (Agilent # G2991AA) as per the manufacturer’s instructions. The library was prepared for sequencing on the NovaSeq6000 (Illumina) using NovaSeq6000 SP kit v1.5, 2 × 150 bp paired-end chemistry, in the Australian Centre for Ecogenomics according to the manufacturer’s instructions. The RNA reads were processed using Trimmomatic 0.4.0 [55] for a minimum length of 150 bp. FastQC 0.12.1 [56] was used to examine the quality of the reads before and after trimming.

### 4.6. Oxford Nanopore Long-Read Sequencing and Data Processing

Sequencing libraries were prepared for the extracted sheep louse gDNA and cross-linked DNA using the Ligation Sequencing Kit SQK-LSK-110 (Oxford Nanopore Technologies, Cambridge, UK) with modifications according to an in-house protocol [42]. The sequencing libraries were each loaded onto a PromethION flowcell (Oxford Nanopore Technologies, Cambridge, UK) for long-read sequencing with the MinKNOW software 22.05.10 on a PromethION sequencer (Oxford Nanopore Technologies, Cambridge, UK). The sequencing process was terminated when approximately 38 Gb of data from the gDNA and 5.5 Gb of data from the cross-linked DNA were generated. The modified base-calling from raw signal data was performed using Guppy 6.4.8 on a high-performance computing cluster.

Two forms of DNA contamination were considered: from the host and from the natural microbiota. The basecalled reads were mapped to the prokaryotic assemblies and *Ovis aries* (GCF_016772045.1) assembly downloaded from RefSeq with Minimap 2.25 [57]. The unmapped reads from the contamination filtering step were recruited for de novo assembly.

### 4.7. De Novo Assembly of Bovicola ovis Genome

To remove the adapters on the raw reads, Porechop 0.2.4 [58] was employed before the reads were filtered with NanoFilt 2.7.0 [59] for reads with quality greater than or equal to quality score 12. The remaining reads were mapped to the *B. ovis* mitochondrial minichromosomes (MH001201.1–MH0012012.1) [60] to separate reads belonging to the *B. ovis* mitochondrial genome and to the *B. ovis* chromosomes. The haploid genome size and heterozygosity were estimated using GenomeScope 2.0 [61]. The genome reads were filtered to remove those shorter than 500 bp and with a read quality below 12 using Nanofilt v2.8.0 [59] before being assembled with Flye 2.9.1 [62]. To improve the quality, Racon 1.4.3 [63] was implemented to polish the draft assemblies using the respective long-read sequences. Purge_Dups 1.2.6 [64] was used on the polished contigs to remove duplicated haplotypes. The completeness and quality of the complete genomes were evaluated using Samtools 1.10 [65], QUAST 5.0.2 [66] and BUSCO 5.4.5 [34] with the long reads sequenced from gDNA. The identity of the complete genomes was verified by BLAST [67].

### 4.8. Chromosome Assembly of B. ovis Using Pore-C Chromatin Interaction Mapping Analysis

The Pore-C data were incorporated for the scaffolding of the draft *B. ovis* genome assembled in this work using the Pore-C analysis pipeline developed by Oxford Nanopore Technologies (ONT) on snakemake workflow [68]. In short, a virtual in silico restriction map with NlaIII was first constructed with the draft assemblies of *B. ovis* using pore_c refgenome virtual digest tool. The Pore-C reads were mapped to the draft genome using BWA 0.7.17 [69] and the alignments were annotated with and assigned to the respective restriction segments based on the restriction map, thus recovering the multi-way contact. To produce the scaffolding, the long-range multi-way contact was decomposed to pairwise contact information, which were then submitted to SALSA 2.3 [70]. The resulting scaffolds with improved assembly continuity were visualised on Juicebox 2.3.0 [71] and curated accordingly. Scaffolds that were smaller than the reported size of the microchromosomes (0.5–1.5 μm) [72] and had yielded no BUSCO genes were removed from the draft assemblies.

### 4.9. Annotation

The structural and functional annotation of the draft assembly of *B. ovis* genome were conducted using the GenSAS 6.0 annotation pipeline [73]. The repetitive elements and non-coding regions were annotated using ab initio prediction with RepeatModeler 2.0.1 [74] and homologous comparison with RepeatMasker 4.1.1 [75]. The interspersed and low complexity repeats were masked with the ab initio database generated from RepeatMasker 4.1.1 [75], and the library of repeat sequences specific to *Drosophila* and *Anopheles gambiae* using NCBI/rmblast search engine in quick and sensitive mode. The draft assembly was masked according to the information generated from both RepeatModeler 2.0.1 [74] and RepeatMasker 4.1.1 [75] prior to feature prediction. Species-specific transcript alignment was performed by aligning the RNA-Seq data generated in this study to the draft genome using HISAT2 2.2.1 [31] and Tophat 2.1.1 [32]. Additionally, the species-related RNA and protein alignments were conducted by mapping the published RefSeq data of *Pediculus humanus corporis* (GCF_000006295.1) and invertebrates using pBLAT 2.5 [28], BLAST+ 2.12.0 [27] and DIAMOND 2.0.6 [29] alignment tools.

The ab initio structural annotations of the draft assembly were predicted by GeneMarkES 4.48 [35], and reference-based structural gene prediction was executed by AUGUSTUS 3.4.0 [30] based on the *Drosophila melanogaster* predefined training set and a customised training set of *P. humanus corporis* (GCF_000006295.1). Additionally, the gene prediction from the RNA alignment results from both HISAT2 [31] and Tophat [32] was performed with BRAKER2 2.1.5 [33], which incorporated Augustus [30] and GeneMark 4.48 [35] in its gene prediction pipeline.

EvidenceModeler 1.1.1 [37] was employed to generate a weighted consensus gene structures from the output generated from ab initio gene predictions and reference-based prediction from protein and transcript alignments (Table 5). OrthoFinder v2.5.5 [76] was used to identify the orthologues shared between the *B. ovis* and *P. humanus corporis* (GCF_000006295.1). Additionally, the unique genes expressed in *B. ovis* but not in *P. humanus corporis* were further investigated manually.

## Figures and Tables

**Figure 1 ijms-25-07824-f001:**
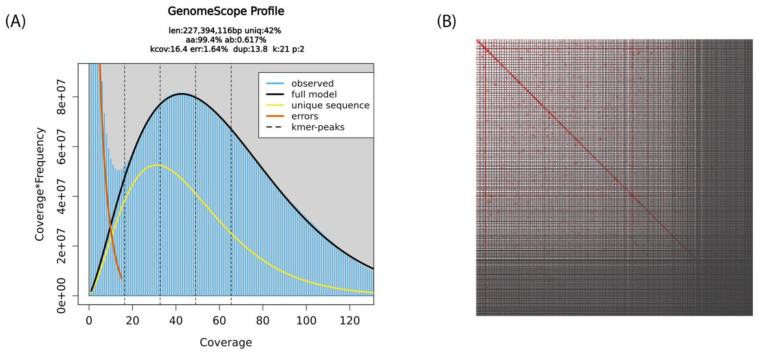
Schematic representation of the *B. ovis* genome statistics: (**A**) Distribution of k-mer frequency in *B. ovis* sequencing data generated with Nanopore sequencing technology at k-mer size of 21. This GenomeScope profile describes the estimated length (len), percent of the genome that is unique (uniq), rate of homozygosity (aa), rate of heterozygosity (ab), mean k-mer coverage (kcov), error rate (err) and average rate of duplication (dup). (**B**) The *B. ovis* genome contig contact matrix using pore-C data. The colour intensity represents the Pore-C contact density in the plot.

**Figure 2 ijms-25-07824-f002:**
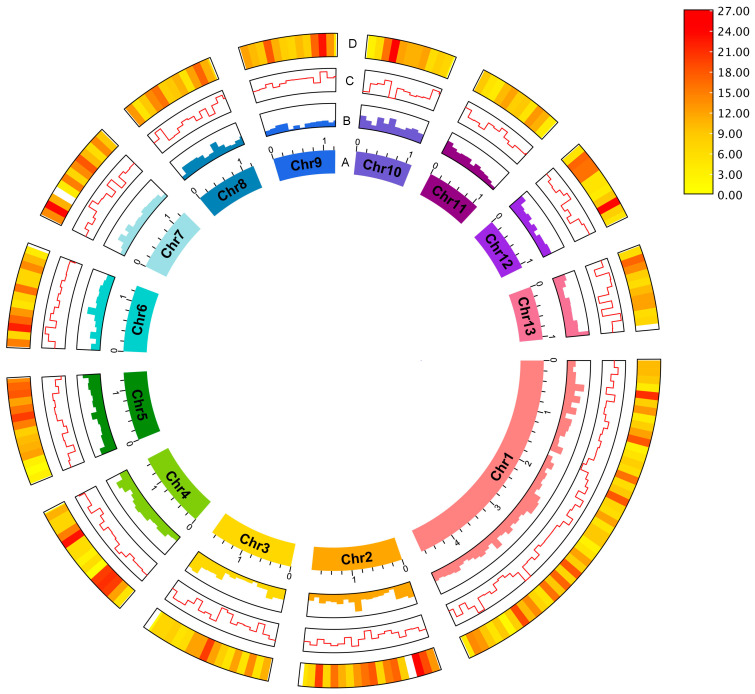
The sheep-biting louse *B. ovis* genome. The circus plot shows the genomic features for 13 pseudo-chromosomes: (A) Pseudo-chromosome, (B) GC content, (C) GC skew and (D) gene density. Gene density was represented by a colour gradient ranging from red to yellow, with higher density indicated by red, and lower density indicated by yellow. The pseudo-chromosome size scale is shown in Mb scale.

**Figure 3 ijms-25-07824-f003:**
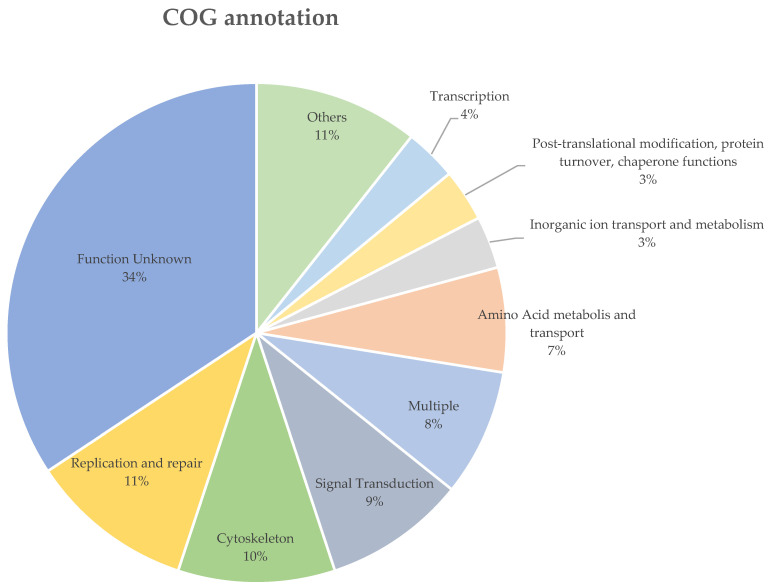
The COG functional annotation of the uniquely expressed genes in *B. ovis*. COG groups with less than 4% were grouped as “Others”, the details of these COG groups are listed in Appendix A.

**Table 1 ijms-25-07824-t001:** Genome size and BUSCO score of several louse species.

Species	Genome Size	Before Scaffolding		After Scaffolding		Reference
		N50	BUSCO Completeness	N50	BUSCO Completeness	
*Brueelia nebulosa*	~114 Mbp	~281 kb	96.40%	~637 Kbp	96.1%	[22]
*Columbicola columbae*	~208 Mbp	~511 kb	NA	17.6 Mbp	96.40%	[23]
*Pediculus humanus*	~110 Mbp	~488 kb	NA	NA	NA	[24]
*Bovicola ovis*	~123 Mbp	~340 kb	96.60%	~416 Kbp	96.6%	This study

**Table 2 ijms-25-07824-t002:** Number of alignments matches identified for *Bovicola ovis* transcripts or polypeptides against different databases using different search engines.

Reference Database	Tool	Evidence Type	Match	Match_Part
GCF_000006295.1	blastn	Transcript	3801	6808
GCF_000006295.1	blastx	Protein	7253	20,277
GCF_000006295.1	BLAT	Transcript	241	382
GCF_000006295.1	DIAMOND	Protein	15,863	15,863
RefSeq invertebrate	blastn	Transcript	110,523	153,379
RefSeq invertebrate	BLAT	Transcript	331,625	712,937
RefSeq invertebrate	DIAMOND	Protein	6,043,377	6,043,377
Swissprot	DIAMOND	Protein	198,696	198,696

**Table 3 ijms-25-07824-t003:** Number of predicted genes, mRNA and polypeptides from the *B. ovis* genome using different gene prediction tools.

	Augustus	Augustus	BRAKER	GeneMarkES	SNAP
Reference	*Drosophila*	GCF_000006295.1	Not Applicable	Not Applicable	*Drosophila*
gene	1612	20,977	11,184	11,313	1438
mRNA	1612	20,977	12,443	11,313	1438
protein	1602	20,943	12,421	11,276	1432
BUSCO score	2.80%	94%	94.70%	93.80%	0.10%

**Table 4 ijms-25-07824-t004:** Other genomic features identified within the *B. ovis* genome.

Genomic Feature	Number	Tool
orf	1,277,282	getorf
ssr	127,654	SSR Finder
rRNA	47	RNAmmer
tRNA	84	tRNAscan-SE

**Table 5 ijms-25-07824-t005:** The weight of each annotation tool in generating the consensus sequences using EvidenceModeler 1.1.1 [37].

Structural Annotation Tool	Reference	Weight
Augustus	GCF_000006295.1	15
BLAST nucleotide (blastn)	GCF_000006295.1	5
BLAST nucleotide (blastn)	NCBI RefSeq invertebrate	10
BLAST proteins (blastx)	GCF_000006295.1	5
BLAT	NCBI RefSeq invertebrate	15
BLAT	GCF_000006295.1	5
BRAKER	RNA alignment	15
DIAMOND proteins	SWISSPROT	5
DIAMOND proteins	GCF_000006295.1	5
DIAMOND proteins RefSeq	NCBI RefSeq invertebrate	10
GeneMarkES	ab initio	5

**Table 6 ijms-25-07824-t006:** Functional annotation for the *B. ovis* genome identified using different tools and against various databases.

Database	Tool	Identified Genes
GCF_000006295.1	blastp	9463 (80.13%)
GCF_000006295.1	Diamond	9042 (76.56%)
InterPro	InterPro	10,090 (85.44%)
Pfam	Pfam	9241 (78.25%)
RefSeq invertebrate	Diamond	9841 (83.33%)
RefSeq invertebrate	blastp	9979 (84.50%)
SignalP	SignalP	11,444 (96.90%)
SwissProt	blastp	7839 (66.38%)
SwissProt	Diamond	6393 (54.13%)
TREMBL	Diamond	9827 (83.21%)

## Data Availability

The datasets generated during the current study are available in the NCBI sequence read archive (SRA) database under BioProject PRJNA1121947 and BioSamples SAMN41769694.

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
