# Peer review of "Chromosome-Scale Genome Assembly of the Sheep-Biting Louse Bovicola ovis Using Nanopore Sequencing Data and Pore-C Analysis"

_ijms, 2024, doi:10.3390/ijms25147824_

Round 1

Reviewer 1 Report

Comments and Suggestions for Authors

This manuscript describes the sequencing and assembly of a genome for B. ovis. This manuscript needs minor revisions to be acceptable for publication. Specific comments are noted below. 

One thought that would be nice but is not required. Rather than the Hi-C versus Pore-C comparison in the discussion, it would be nice to have spent the space discussing something novel about the genome that came from the annotation or RNA-Seq. You discuss resistance as a problem in this organism. Did you expansion in the cytochrome P450s or other esterases? 

Please review for proper usage of verb tenses. Example L47, "reliance on chemicals is makes..."

L53-55: This sentence seems to need a reference as it references specific facts.

L134: Italicize species name

L142: Do you have a misplaced decimal point? Is the genome 1.23Mbp or 123 Mbp?

L195: Is it odd that the number of mRNA exactly matches the number of genes? As one of many examples, most insect sodium channel genes have more than 20 transcripts due to alternative splicing. Wouldn't you expect the number of mRNA to be higher than the number of genes? A 1:1 correspondence seems exceptional.

L211: sheep louse

L231-251: What is the point of these two paragraphs in the midst of a genome manuscript? It seems out of place to do a couple of paragraph comparison to Hi-C, a method not used in this study. 

L270-272: Please revise this sentence to improve readability.

L305: Protein degradation "reaction?" Do you mean reagent or solution?

L316: Missing values. Please correct.

L355: This phrase needs revision. I think you mean "eliminate reads shorter than and with a quality score below 12." As currently written, the highest quality reads (>12) were removed.

Also, what tool and what data were used for this filtering- was this filtering done on actual reads or was this done on the nanopore sequencing summary file which contains the metadata (including length and read quality numbers)?

L365: "in silico" is traditionally italicized

Comments on the Quality of English Language

English needs some revision to improve flow and for standard usage. A couple of examples are noted in the Comments to Authors.

Author Response

Queensland Alliance for Agriculture and Food Innovation (QAAFI)

1 July 2024

MDPI IJMS Editorial Office

Dear Reviewer 1,

RE: Revision of Manuscript ID ijms-3075346 entitled “Chromosome-scale genome assembly of the sheep biting louse Bovicola ovis using Nanopore sequencing data and Pore-C analysis”

We would like to extend our heartfelt appreciation for the valuable feedback provided by the reviewer on our manuscript. Their insightful comments and suggestions have been instrumental in guiding us to improve our work. We have carefully considered and incorporated their feedback, which has significantly enhanced the quality of our manuscript.

We thank the reviewer once again for their time and effort in reviewing our submission. Their contributions have been invaluable to us.

Responses to Reviewer 1

*The reviewer’s comments are highlighted in yellow.

This manuscript describes the sequencing and assembly of a genome for B. ovis. This manuscript needs minor revisions to be acceptable for publication. Specific comments are noted below. 

One thought that would be nice but is not required. Rather than the Hi-C versus Pore-C comparison in the discussion, it would be nice to have spent the space discussing something novel about the genome that came from the annotation or RNA-Seq. You discuss resistance as a problem in this organism. Did you expansion in the cytochrome P450s or other esterases? 

Dear reviewer, thank you for your suggestion. We modified the paragraph, which described the Pore-C strength in achieving a high-quality assembly with limited sample tissue in our case, as per following to highlight our main message to deliver in those paragraphs.

Also, we added some analyses and discussion regarding the gene annotation and the similarity between B. ovis and P. humanus corporis.

Methods:

OrthoFinder v2.5.5 [72] was used to identify the orthologues shared between the B. ovis and P. humanus corporis (GCF_000006295.1). The unique genes expressed in B. ovis but not in P. humanus corporis were further investigated manually.

Results:

The B. ovis sequenced in this study and the P. humanus corporis (GCF_000006295.1) shared 99.1% and 98.4% of their orthogroups with each other respectively. There were 541 genes (4.6%), which were grouped in 121 orthogroups, uniquely expressed in B. ovis. In total, only 195 of the 541 uniquely expressed genes in B. ovis were assigned to a COG functional group, of which 71 (36.4%) was labelled as “Function unknown” (Supplementary File 2). The most abundant COG group with known function was “Replication and repair” (n=22), followed by and “Cytoskeleton” (n=21), “Signal Transduction” (n=19) (Figure 3. Additionally, 167 of the unique genes retuned no blast results and 170 were labelled as either hypothetical, unnamed protein or uncharacterized proteins (Supplementary File 1).

Figure 3. The COG functional annotation of the uniquely expressed genes in B. ovis.

Discussion:

Our analysis revealed a relatively high similarity between B. ovis and P. humanus corporis (GCF_000006295.1) as 99.1% of the gene orthologs in B. ovis were also reported in P. humanus corporis (GCF_000006295.1). Further investigation of the unique genes in B. ovis reported several key genes that may be linked to pesticide resistance mechanisms in B. ovis. For instance, two cytochrome b-c1 complex subunit 8-like genes were identified, which are crucial for cellular respiration and energy production. These cytochromes might play significant roles in metabolic adaptations and resistance [49]. Several other genes that can have potential roles in resistance mechanisms: an ABC transporter (putative), known for mediating multidrug resistance [50] and acetylcholine receptor protein subunit alpha-L1 precurso(putative), potentially linked to neurotoxin resistance [51]; a dual specificity protein phosphatase CDC14 (putative) involved in cell cycle regulation [52]; and glucose dehydrogenase (putative) impacting energy metabolism [53] were identified. Furthermore, identified genes like the ionotropic glutamate receptor, histone-lysine N-methyltransferase SETMAR, and Rho-associated protein kinase can also further our understanding of resistance to insecticides. Overall, these findings provide valuable insights into the molecular mechanisms of B. ovis resistance against control treatments, aiding the development of effective control strategies for the sheep industry.

Please review for proper usage of verb tenses. Example L47, "reliance on chemicals is makes..."

We have removed the “is” from the sentence.

L53-55: This sentence seems to need a reference as it references specific facts.

We have added the missing reference to the end of the sentence. The cited reference is as below:

  1. James PJ, Callander JT. Dipping and jetting with tea tree (Melaleuca alternifolia) oil formulations control lice (Bovicola ovis) on sheep. Vet Parasitol. 2012;189(2):338-43.

L134: Italicize species name

We have italicized the species name.

L142: Do you have a misplaced decimal point? Is the genome 1.23Mbp or 123 Mbp?

Yes we misplaced the decimal point. We have corrected it to 123 Mbp.

L195: Is it odd that the number of mRNA exactly matches the number of genes? As one of many examples, most insect sodium channel genes have more than 20 transcripts due to alternative splicing. Wouldn't you expect the number of mRNA to be higher than the number of genes? A 1:1 correspondence seems exceptional.

Dear reviewer, the numbers reported in this part are the predicted number of genes, mRNA and proteins using multiple tools and weighted with PASA. The actual transcripts sequenced in this study were used later to confirm and annotate these predicted genes.

We improved the sentence to hopefully increase the clarity.

PASA [38] was employed to weigh and refine the predicted gene set using the cDNA transcripts of P. humanus corporis (GCF_000006295.1), which resulted with 11,810 genes, 11,810 mRNA and 11,444 proteins in total.

L211: sheep louse

We have corrected from sheep lice to sheep louse.

L231-251: What is the point of these two paragraphs in the midst of a genome manuscript? It seems out of place to do a couple of paragraph comparison to Hi-C, a method not used in this study. 

Dear reviewer, the paragraph was intended to explain the usage and the benefits of Pore-C data particularly when our starting sample material was limited. We modified the paragraph as per following to highlight the good news of achieving a high-quality assembly with limited material.

“In this study, we reported a chromosome-scale genome assembly of the sheep biting louse, B. ovis, using Nanopore sequencing data and Pore-C analysis. Pore-C captures a comparable number of chromosomal interactions to Hi-C with less sequencing data, demonstrating high efficiency [46-48]. Our chromatin conformation analysis showed 58.17% two-way interactions and 41.83% multi-way interactions, similar to human cells (47.56%) [48] and Arabidopsis (44%) [47]. Longer reads with more contacts provide richer multi-dimensional interaction data, enhancing the quality of de novo genome assembly. Although only one Pore-C library was generated due to limited tissue, the high consistency in Pore-C data from Arabidopsis [47], suggests the data here is of sufficient quality for assembly. This highlights Pore-C's reliability and effectiveness, even with limited samples, justifying its use in complex genomic studies.”

L270-272: Please revise this sentence to improve readability.

We have revised this sentence as per following:

“Wool samples from sheep suspected of louse infestation were generously provided by local Queensland sheep farms.”

L305: Protein degradation "reaction?" Do you mean reagent or solution?

It is a “reaction” according to the manufacturer’s protocol. however, we added the details of the reaction to avoid confusion. Below is the updated version of the sentence:

The restriction enzyme digestion was subsequently inactivated before the proximity ligation reaction (5% Tween-20, 0.5% SDS and proteinase K) was added to the mixture.

L316: Missing values. Please correct.

We have updated this sentence.

L355: This phrase needs revision. I think you mean "eliminate reads shorter than and with a quality score below 12." As currently written, the highest quality reads (>12) were removed.

Also, what tool and what data were used for this filtering- was this filtering done on actual reads or was this done on the nanopore sequencing summary file which contains the metadata (including length and read quality numbers)?

Thank you for pointing it out! We have corrected the sentence as per following:

The genome reads were filtered to remove those shorter than 500 bp and with a read quality below 12 using Nanofilt v2.8.0 [55] before being assembled with Flye 2.9.1 [58].

L365: "in silico" is traditionally italicized

We have italicized the word as per suggested.

Comments on the Quality of English Language

English needs some revision to improve flow and for standard usage. A couple of examples are noted in the Comments to Authors.

We have asked a few native English speaker to peer review the latest version of our manuscript and have improved the manuscript with their combined input.

Again, we thank the reviewer for their valuable comments which have helped us to improve the manuscript. We sincerely hope these edits are acceptable. Thank you!

Yours sincerely

Dr. Chian Teng Ong

QAAFI Early Career Research Fellow

Centre for Animal Science

Queensland Alliance for Agriculture and Food Innovation (QAAFI), UQ Australia

Reviewer 2 Report

Comments and Suggestions for Authors

The study "Chromosome-scale genome assembly of the sheep biting louse Bovicola ovis using Nanopore sequencing data and Pore-C analysis" presents the first genome assembly and annotation of B. ovis using ling reads. It is the first genome sequenced and annotated of the Genus and it will be a useful tool for the development of control tools against the lice B. ovis that affects the sheep in Australia. 

The authors have done a nice work in this genome announcement paper and I believe it will be greatly appreciated from colleagues in this field since it will provide important information regarding the genome of this species. Therefore, I recommend it for publication after some minor editing.

L28: add ",respectively" at the end of the sentence. 

L33. Use the whole name of the species when at the beginning of the sentence

L44: it should read "identify"

L47: delete "is"

L63-65: This sentence lacks a main verb and therefore, it does not make sense. please revise accordingly.

L91-92: delete "in improve the"

Figure 1: Use the abbreviation of the species B. ovis and always in italics

L138: Put a full stop after performed and start then a new sentence 

L145: replace "as" with "at"

Figure 2: use the abbreviation of the species. Alos what does the colours in gene density mean? provide explanation. 

Table 1: some species and genome size numbers are in bold and some not. make it uniform.

L155: use A. gambiae

L236-237: This sentence lacks a main verb and therefore, it does not make sense. please revise accordingly.

It would also be useful to discuss about the gene families that were discovered during annotation. what was the conservation level detected between B. ovis and Pediculus humanus?

Comments on the Quality of English Language

English syntax requires minor editing

Author Response

Queensland Alliance for Agriculture and Food Innovation (QAAFI)

1 July 2024

MDPI IJMS Editorial Office

Dear Reviewer 2,

RE: Revision of Manuscript ID ijms-3075346 entitled “Chromosome-scale genome assembly of the sheep biting louse Bovicola ovis using Nanopore sequencing data and Pore-C analysis”

We would like to extend our heartfelt appreciation for the valuable feedback provided by the reviewer on our manuscript. Their insightful comments and suggestions have been instrumental in guiding us to improve our work. We have carefully considered and incorporated their feedback, which has significantly enhanced the quality of our manuscript.

We thank the reviewer once again for their time and effort in reviewing our submission. Their contributions have been invaluable to us.

Responses to Reviewer 2

*The reviewer’s comments are highlighted in yellow.

The study "Chromosome-scale genome assembly of the sheep biting louse Bovicola ovis using Nanopore sequencing data and Pore-C analysis" presents the first genome assembly and annotation of B. ovis using ling reads. It is the first genome sequenced and annotated of the Genus and it will be a useful tool for the development of control tools against the lice B. ovis that affects the sheep in Australia. 

The authors have done a nice work in this genome announcement paper and I believe it will be greatly appreciated from colleagues in this field since it will provide important information regarding the genome of this species. Therefore, I recommend it for publication after some minor editing.

L28: add ",respectively" at the end of the sentence. 

We corrected the sentence accordingly.

L33. Use the whole name of the species when at the beginning of the sentence

We corrected the name of the species accordingly.

L44: it should read "identify"

We have corrected the word.

L47: delete "is"

We have removed the “is” from the sentence.

L63-65: This sentence lacks a main verb and therefore, it does not make sense. please revise accordingly.

We have revised the sentence as per following:

Since 2020, various molecular methods, including Polymerase Chain Reaction (PCR), looped-mediated isothermal amplification (LAMP) and quantitative PCR (qPCR), have been used for sheep louse detection [7, 15]. 

L91-92: delete "in improve the"

We deleted the words as per suggested.

Figure 1: Use the abbreviation of the species B. ovis and always in italics

We corrected the species name as per suggested

L138: Put a full stop after performed and start then a new sentence 

We split the sentence as per suggested

L145: replace "as" with "at"

We replaced “as” with “at” as per suggested.

Figure 2: use the abbreviation of the species. Also what does the colours in gene density mean? provide explanation. 

We corrected the species name. We added a legend and some description in the figure legend to explain the colours in the gene density.

Table 1: some species and genome size numbers are in bold and some not. make it uniform.

We have fixied the inconsistency in Table 1.

L155: use A. gambiae

We corrected the species name

L236-237: This sentence lacks a main verb and therefore, it does not make sense. please revise accordingly.

We corrected the sentence as per following:

The number of fragments associated with the set of Pore-C contacts is referred to as the contact order [46].

It would also be useful to discuss about the gene families that were discovered during annotation. what was the conservation level detected between B. ovis and Pediculus humanus?

Dear reviewer, thank you for your suggestion. We added some analyses and discussion regarding the gene annotation and the similarity between B. ovis and P. humanus corporis.

Methods:

OrthoFinder v2.5.5 [72] was used to identify the orthologues shared between the B. ovis and P. humanus corporis (GCF_000006295.1). The unique genes expressed in B. ovis but not in P. humanus corporis were further investigated manually.

Results:

The B. ovis sequenced in this study and the P. humanus corporis (GCF_000006295.1) shared 99.1% and 98.4% of their orthogroups with each other respectively. There were 541 genes (4.6%), which were grouped in 121 orthogroups, uniquely expressed in B. ovis. In total, only 195 of the 541 uniquely expressed genes in B. ovis were assigned to a COG functional group, of which 71 (36.4%) was labelled as “Function unknown” (Supplementary File 2). The most abundant COG group with known function was “Replication and repair” (n=22), followed by and “Cytoskeleton” (n=21), “Signal Transduction” (n=19) (Figure 3. Additionally, 167 of the unique genes retuned no blast results and 170 were labelled as either hypothetical, unnamed protein or uncharacterized proteins (Supplementary File 1).

Figure 3. The COG functional annotation of the uniquely expressed genes in B. ovis.

Discussion:

Our analysis revealed a relatively high similarity between B. ovis and P. humanus corporis (GCF_000006295.1) as 99.1% of the gene orthologs in B. ovis were also reported in P. humanus corporis (GCF_000006295.1). Further investigation of the unique genes in B. ovis reported several key genes that may be linked to pesticide resistance mechanisms in B. ovis. For instance, two cytochrome b-c1 complex subunit 8-like genes were identified, which are crucial for cellular respiration and energy production. These cytochromes might play significant roles in metabolic adaptations and resistance [49]. Several other genes that can have potential roles in resistance mechanisms: an ABC transporter (putative), known for mediating multidrug resistance [50] and acetylcholine receptor protein subunit alpha-L1 precurso(putative), potentially linked to neurotoxin resistance [51]; a dual specificity protein phosphatase CDC14 (putative) involved in cell cycle regulation [52]; and glucose dehydrogenase (putative) impacting energy metabolism [53] were identified. Furthermore, identified genes like the ionotropic glutamate receptor, histone-lysine N-methyltransferase SETMAR, and Rho-associated protein kinase can also further our understanding of resistance to insecticides. Overall, these findings provide valuable insights into the molecular mechanisms of B. ovis resistance against control treatments, aiding the development of effective control strategies for the sheep industry.

Comments on the Quality of English Language

English syntax requires minor editing

We have asked a few native English speaker to peer review the latest version of our manuscript and have improved the manuscript with their combined input.

Again, we thank the reviewer for their valuable comments which have helped us to improve the manuscript. We sincerely hope these edits are acceptable. Thank you!

Yours sincerely

Dr. Chian Teng Ong

QAAFI Early Career Research Fellow

Centre for Animal Science

Queensland Alliance for Agriculture and Food Innovation (QAAFI), UQ Australia
